# Increased expression of Netrin-4 is associated with allodynia in a trigeminal neuropathic pain model rats by infraorbital nerve injury

Yuka Honjo[1,2], Yuki Fujita[1,3], Hitoshi Niwa[2], Toshihide Yamashita[1,3,4,5] *

**1** Department of Molecular Neuroscience, Osaka University Graduate School of Medicine, Suita, Osaka, Japan, **2** Department of Dental Anesthesiology, Osaka University Graduate School of Dentistry, Suita, Osaka, Japan, **3** Immunology Frontier Research Center, Osaka University, Suita, Osaka, Japan, **4** Department of Molecular Neuroscience, Osaka University, Graduate School of Frontier Biosciences, Osaka University, Suita, Osaka, Japan, **5** Department of Neuro-Medical Science, Graduate School of Medicine, Osaka University, Osaka, Japan

* yamashita@molneu.med.osaka-u.ac.jp

**Data Availability Statement:** All relevant data are within the paper and its Supporting Information files.

## Abstract

Neuropathic pain refers to pain caused by lesions or diseases of the somatosensory nervous system that is characteristically different from nociceptive pain. Moreover, neuropathic pain occurs in the maxillofacial region due to various factors and is treated using tricyclic antidepressants and nerve block therapy; however, some cases do not fully recover. Netrin is a secreted protein crucially involved in neural circuit formation during development, including cell migration, cell death, neurite formation, and synapse formation. Recent studies show Netrin-4 expressed in the dorsal horn of the spinal cord is associated with chronic pain. Here we found involvement of Netrin-4 in neuropathic pain in the maxillofacial region. Netrin-4, along with one of its receptors, Unc5B, are expressed in the caudal subnucleus of the trigeminal spinal tract nucleus. Inhibition of its binding by anti-Netrin-4 antibodies not only shows a behavioral analgesic effect but also neuronal activity suppression. There was increased Netrin-4 expression at 14 days after infraorbital nerve injury. Our findings suggest that Netrin-4 induced by peripheral nerve injury causes neuropathic pain via Unc5B.

## Introduction

Neuropathic pain is defined as "pain caused by a lesion or disease of the somatosensory system" by the International Association for the Study of Pain [1] and is characteristically different from nociceptive pain. It involves spontaneous sensations (persistent or intermittent) and stimulation-induced pain (allodynia and hyperalgesia) in areas coinciding with the impaired innervation area, as well as various sensations induced by nerve damage. Moreover, there is a characteristic of converging abnormalities [2]. It might also involve the trigeminal nerve region as a result of various factors, including surgery-induced trauma, tumors, neurodegenerative diseases, etc. Specifically, during dental treatments, nerves are inevitably mechanically cut during teeth extraction; moreover, there could be accidental damage to the lingual nerve

**Funding:** This work was supported by Japan Agency for Medical Research and Development (18gm1210005h0001) to T.Y. https://www.amed. go.jp/en/index.html The funders had no role in study design, data collection and analysis, decision to publish, or preparation of the manuscript.

**Competing interests:** The authors have declared that no competing interests exist.

and inferior alveolar nerve during oral surgery. Consequently, there is a high possibility of neuropathic pain caused by nerve damage [3]. Its pharmacological treatments include tricyclic antidepressants, selective norepinephrine and serotonin reuptake inhibitors, opioid agonists, and nerve block treatments. However, there are some cases of insufficient analgesic effect or where treatment cannot be used due to age or side effects [4]. Therefore, there is a need to establish an effective treatment by elucidating a novel molecular mechanism.

Netrin, which is a laminin-related extracellular protein, was originally identified as an attractant molecule for axon guidance in the embryonic spinal cord [5, 6]. In mammals, there are four secreted types (Netrin-1, -3, -4, and -5) and two glycosylphosphatidylinositol-anchored types (Netrin-G1 and -G2) [6–8]. Secreted Netrins bind to DCC (deleted in colorectal cancer), Neogenin, and Unc5 (A-D) while Netrin-Gs bind to different proteins [9]. Previous studies have demonstrated the important roles of Netrin in cell migration, survival, axon branching, and synaptogenesis during neural development [10–13]. The role of Netrin during adulthood has remained unclear; however, there have been recent reports regarding its role.

Netrin-4 is very homologous to the laminin β1 chain. In adults, Netrin-4 is expressed in the lung, kidney, blood vessel, and most brain parts, including the upper trigeminal spinal tract nucleus [14]. Further, Netrin-4 binds to Unc5B in the spinal cord [13, 15]. We previously reported that Netrin-4 delivered by dorsal horn interneurons was involved in inflammatory and neuropathic pain and that inhibition of Netrin-4 binding to Unc5B suppressed post-nerve-injury neuropathic pain [16].

This study aimed to investigate the role of Netrin-4 in orofacial neuropathic pain. Specifically, we aimed to examine Netrin-4 and Unc5B expression in the trigeminal subnucleus caudalis (Vc) of adult rats after infraorbital nerve injury, as well as Netrin-4 effects on mechanical sensitivity using anti-Netrin-4 antibody administration in a chronic pain model.

## Materials and methods

### Animals

6-9-wk-old male Sprague-Dawley rats (190–300 g; Japan SLC) were used in this study. All experiments were accordance with the Osaka University Medical School Guide for the Care and Use of Laboratory Animals and were approved by Institute Committee of Osaka University (Permit Number: 28–058).

### Models for orofacial neuropathic pain

Chronic constriction injury to rat's infraorbital nerve (CCI-ION) was performed with some changes as has been previously described [17]. Rats were put under isoflurane anesthesia. A small incision was made at the edge of the left whisker pad, and left ION was exposed. Two nylon sutures (6–0 thread) were placed 2mm apart around the ION, and the wound was sutured after ligation of the ION. These ligations reduced the diameter of the nerve, but did not occlude it completely (CCI: ipsilateral side). And contralateral ION was exposed but did not undergo nerve ligation for sham injury (Sham: contralateral side). Hereinafter it is described as 'CCI--Sham injury'. We used this model for all subsequent experiments, and the total number of rats which were analyzed in this study were 86 (make animal models; n = 23, administration of antibody; n = 27, immunohistochemical staining; n = 20, quantitative PCR; n = 16).

### Behavioral testing

The mechanical sensitivity of animals was assessed by using the von Frey filament test, as previously described [18]. Rats were placed on an acrylic box (30×30×30 cm) and habituated for

5–10 min before any stimulation. Then mechanical stimulations (4,8,15 and 26 g) were applied to center of the whisker pad using von Fry filaments (Semmes-Weinstein Von Frey esthesiometer, Muromachi Kikai Co.). The head-withdrawal threshold of the mechanical stimulation on the whisker pad skin was defined as minimum pressure needs to evoke at least three reactions by five stimuli (positive reaction; withdrawal of the head, or an attack or escape). The cutoff to prevent tissue damage was determined as 26 g. Before CCI-sham injury, stimulation with 26 g filament did not induce any nociceptive reaction. The stimuli were applied before injury and 7, 14, 21 days after injury.

## Intracisternal administration of drugs

At 14 days after CCI-Sham injury, rats were tested their behavior before administration and put under isoflurane anesthesia. Anesthesia rats were placed in a head holder. A skin incision was made from the back of calvarium to top of cervical spine. The atlanto-occipital membrane was dissected under a surgical microscope, and a micro syringe with a 29-gauge needle inserted into cisterna magna. A mouse anti-Netrin-4 antibody (AF1132, R&D Systems) or Normal Goat IgG (144–09531, Wako) was diluted to 30 µg with 50µl saline and infusions of drags were performed slowly. The skin and muscle wound were sutured rapidly.

## Immunohistochemistry

Immunohistochemistry was performed as described previously (Hayano et al., 2016). 30 minutes after the behavior test, rats were deeply anesthetized and then transcardially perfused with 100ml PBS, followed by 200ml of ice-cold 4% paraformaldehyde (PFA) in a 0.1M phosphate buffer. Vc and upper cervical cord (C1-C2) were dissected and post-fixed in 4% PFA for 24 hours at 4˚C. Tissues were transferred and stored in 30% sucrose in PBS 48 hours at 4˚C. 30µm coronal sections were cut on a cryostat. Sections washed three times and incubated in blocking solution (5% BSA and 0.3% Triton X-100 in PBS) for 1 hour at room temperature. For double-labelling immunofluorescence histochemistry, sections were incubated for 24 or 48 h at 4˚C with antibodies against Netrin-4 (1:50; HPA049832; ATLAS), Unc5B (1:200; HPA01141, ATLAS), NeuN (1:1000; MAB377, Millipore), Phospho-p44/42 MAPK (Erk1/2) (Thr202/Tyr204) (1:200; #4370, CST), Iba1(1:200; ab5076, abcam), GFAP (1:200; G-A-5, Sigma-Aldrich). The sections were washed 3 times and incubated for 2 h at room temperature or overnight at 4˚C with secondary antibodies (Alexa Fluor 488/568/594; 1:500; Thermo Fisher Scientific, Inc.). The sections were washed 3 times, slide-mounted, and cover-slipped in mounting medium with the nuclear stain 4´,6-diamidino-2-phenylindole (DAPI, Vector Laboratories).

The number of pERK positive neurons in the all area of Vc and the laminae II of C1 was counted. The mean number of pERK-immunoreactive (pERK-IR) neurons was calculated from 6 tissue sections (Positions: -2160, -1440, -720, 0, +720 and +1440 µm from the obex).

## Quantitative PCR

Total RNA was extracted from the Vc-C1 which was collected by Laser Microdissection technique (LMD7000, Leica) with TRIzol reagent (15596018, Invitrogen). After cDNA synthesis (High Capacity cDNA RT Kit, Applied Biosystems), gene expression was quantified with a TaqMan Gene Expression Assay (Applied Biosystems). The following Taqman Assays were used: *Netrin-4* (Rn01760394_m1); *Unc5B* (Rn00573551_m1). Cycle threshold (Ct) values were calculated using the ΔΔCt method. All ΔCt values were normalized to rat glyceraldehyde-3-phosphate dehydrogenase (Rn99999916_s1)

### Experimental design and statistical analysis

Statistical analysis was performed using the Mann-Whitney test for comparisons between two groups. For comparisons among multiple groups, the two-way Repeated-Measure ANOVA with the Bonferroni *post-hoc* test and 3-way ANOVA with the Tukey *post-hoc* tests or the Dunnett *post-hoc tests* were used. Differences with P <0.05 were considered statistically significant. Statistical analyses were performed using Prism software (GraphPad software In.). All data are presented as the mean ± standard error of the mean.

## Results

### Development of hyperalgesia after CCI-Sham injury

The CCI-ION rat has been studied as a model of orofacial neuropathic pain [17, 19, 20]. To assess the pain-related behavior of the orofacial neuropathic pain rat model, we measured their head-withdrawal threshold to the whisker pad skin using the von Frey filament test. There was a significant decrease in the head-withdrawal threshold for mechanical stimulation of the ipsilateral side at 14 and 21 days after injury compared with the pre-injury day (Fig 1). There was no significant change in the contralateral side on any day.

### Netrin-4 antibody attenuates pain-related behavior in a trigeminal neuropathic pain rat model

We directly administered anti-Netrin-4 antibody to the cisterna magna to examine whether anti-Netrin-4 antibody, which has an analgesic effect on neuropathic pain caused by sciatic

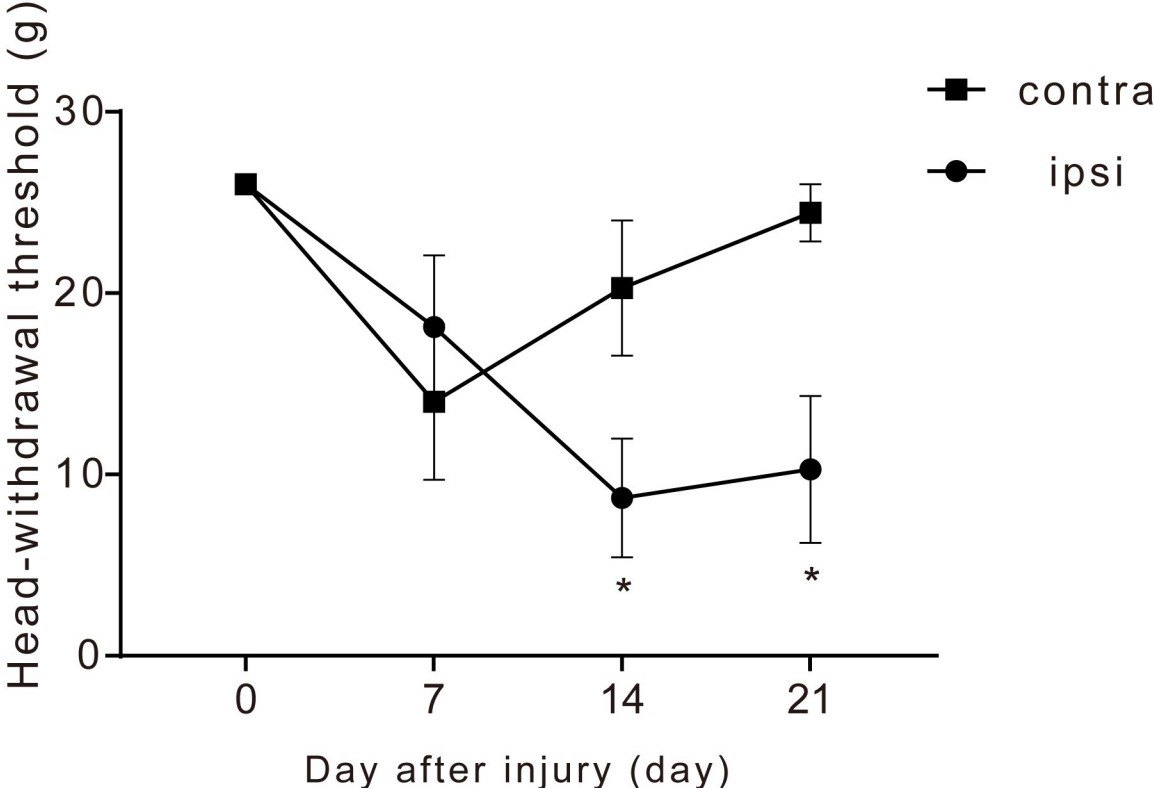

**Fig 1. Time course of the development of hyperalgesia after chronic constriction injury (CCI) of infraorbital nerve (ION).** Head-withdrawal thresholds after CCI-Sham injury. The ipsilateral side shows significantly decrease compared to contralateral side at 14 and 21 day after injury (n = 7). The data are presented as the mean ± SEM.*, P<0.05; Bonferroni test (contra; contralateral, ipsi; ipsilateral).

nerve injury [16], exerted an analgesic effect on neuropathic pain caused by infraorbital nerve injury.

First, Evans Blue was administered to confirm whether the antibody added from the cisterna magna reached the target Vc-C1 region (Fig 2A). Second, a fluorescent-labeled anti-Netrin-4 antibody was administered from the cisterna magna and analyzed to confirm whether it would bind to neuronal cells in the Vc (Lightning-Link Rapid Conjugation System DyLight488, Innova Biosciences), which was confirmed (Fig 2B and 2C).

After confirming that the antibody reached the target region after administration through direct puncture of the cisterna magna, we assessed changes in the head-withdrawal threshold after antibody administration. The anti-Netrin-4 or Control antibodies were administrated at 14 days after CCI-Sham injury. Compared with the Control antibody group, the anti-Netrin-4 antibody group showed a significant increase in the head-withdrawal threshold of the ipsilateral side after one day of administration (at 15 days after CCI-Sham injury) (Fig 2D).

### Suppression of the neuronal activation by anti-Netrin-4 antibody in the Vc

A previous study [16] reported that neurons were activated by Netrin-4 administration to the spinal cord. Therefore, we performed immunohistochemistry to determine the effect of anti-Netrin-4 antibody on neuronal activation in the Vc.

After the 14 treatment days where the head-withdrawal threshold was significantly altered by the CCI-Sham injury, there was the expression of anti-pERK antibody-positive cells on both the ipsilateral and contralateral sides (Fig 3A and 3B). There was a significant increase in the number of cells positive for both the anti-pERK and anti-NeuN antibodies on the ipsilateral side. However, we could not observe the pERK signals in Iba-1-positive cells (Fig 3C) or GFAP-positive cells (Fig 3D). Further, upon anti-Netrin-4 antibody administration, there was a significant decrease in the number of pERK-positive neurons on the ipsilateral side compared with the pre-administration values and those in the control antibody administration group (Fig 3E and 3F).

### Post-CCI increased Netrin-4 expression in the Vc

Previous studies have reported Netrin-4 and Unc5B expression in a healthy adult human brain; specifically, Netrin-4 expression in the trigeminal spinal subnucleus interpolaris (Vi) [14]. Consequently, we hypothesized that there was Netin-4 and Unc5B expression in the Vc. Immunohistochemical examination revealed anti-Netrin-4 and anti-Unc5B antibody-positive cells in the Vc. (Fig 4A–4F). Moreover, there was neuronal expression of Netrin-4 and Unc5B in the Vc (Fig 4D and 4F). Furthermore, to examine whether there was an association of changes in the head-withdrawal threshold with those in Netrin-4 and Unc5B expression in the Vc, we performed quantitative mRNA analysis by real-time PCR. Compared with pre-injury levels, there was a significant increase in the Netrin-4, but not Unc5b, levels from 7 to 21 days on both sides in the Vc after CCI (Fig 4G and 4H). We examined expression of Netrin-4 and Unc5B at 0 and 14 days (Fig 5A, 5B, 5E and 5F), and after anti-Netrin-4 antibody or control antibody treatment (Fig 5C, 5D, 5G and 5H). We did not observe the difference of the expression of Netrin-4 between ipsilateral and contralateral side in each group examined (Fig 5A–5H).

## Discussion

### The model of orofacial neuropathic pain established through infraorbital nerve injury

In this study, we used a rat model established through CCI-ION to analyze the effect of the anti-Netrin-4 antibody on trigeminal neuropathic pain. This model has been used in

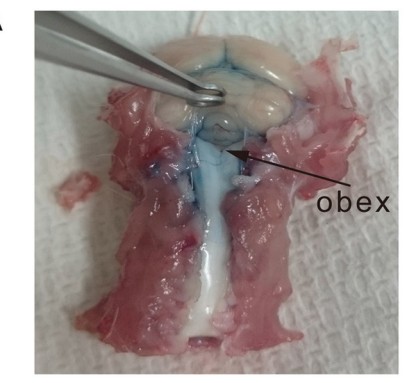

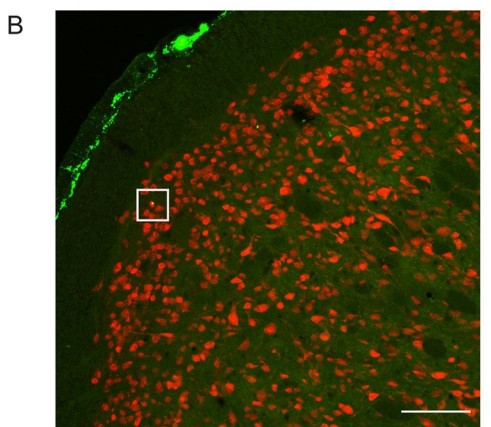

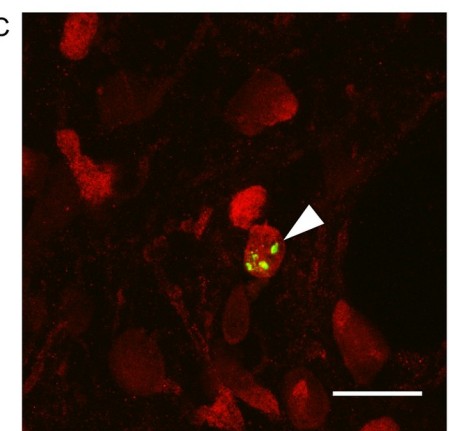

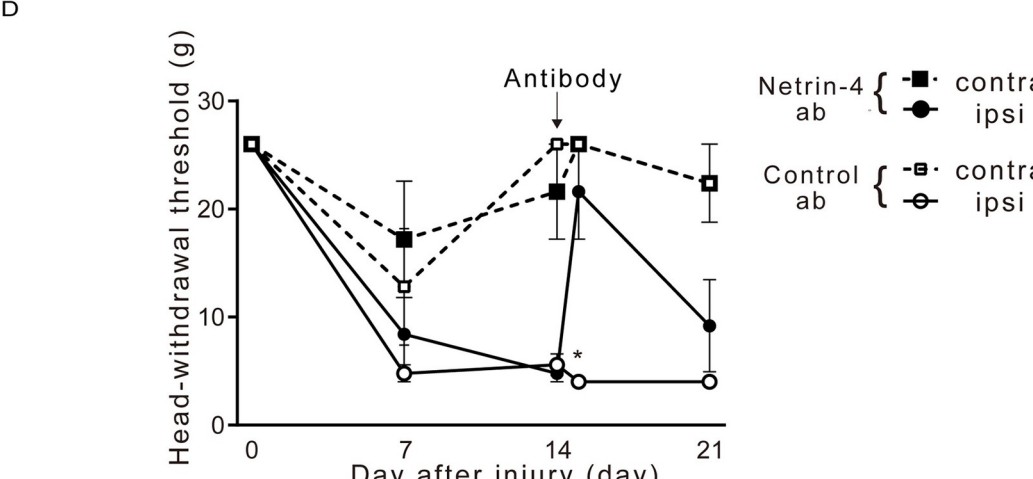

**Fig 2. Effects of anti-Netrin-4 antibody administration on pain behavior in trigeminal neuropathic pain model rats.**
(A) Confirmation of administration area by Evans Blue. (B and C) Penetration of anti-Netrin-4 antibody (Antibody) into the Vc. Double immunofluorescence staining for the fluorescence dye-conjugated Netrin-4 antibody (DyLight 488; green) and NeuN (red). The fluorescence of antibody was observed on neurons (arrowhead) in Vc. (C) is high magnification image corresponding to the square area in (B). Bars: (B) 100 μm, (C) 20 μm. (D) Head-withdrawal threshold after administration control IgG or anti-Netrin-4 antibody to CCI-ION or Sham injury (n = 5). The data are presented as the mean ± SEM. *, P<0.05; Tukey test.

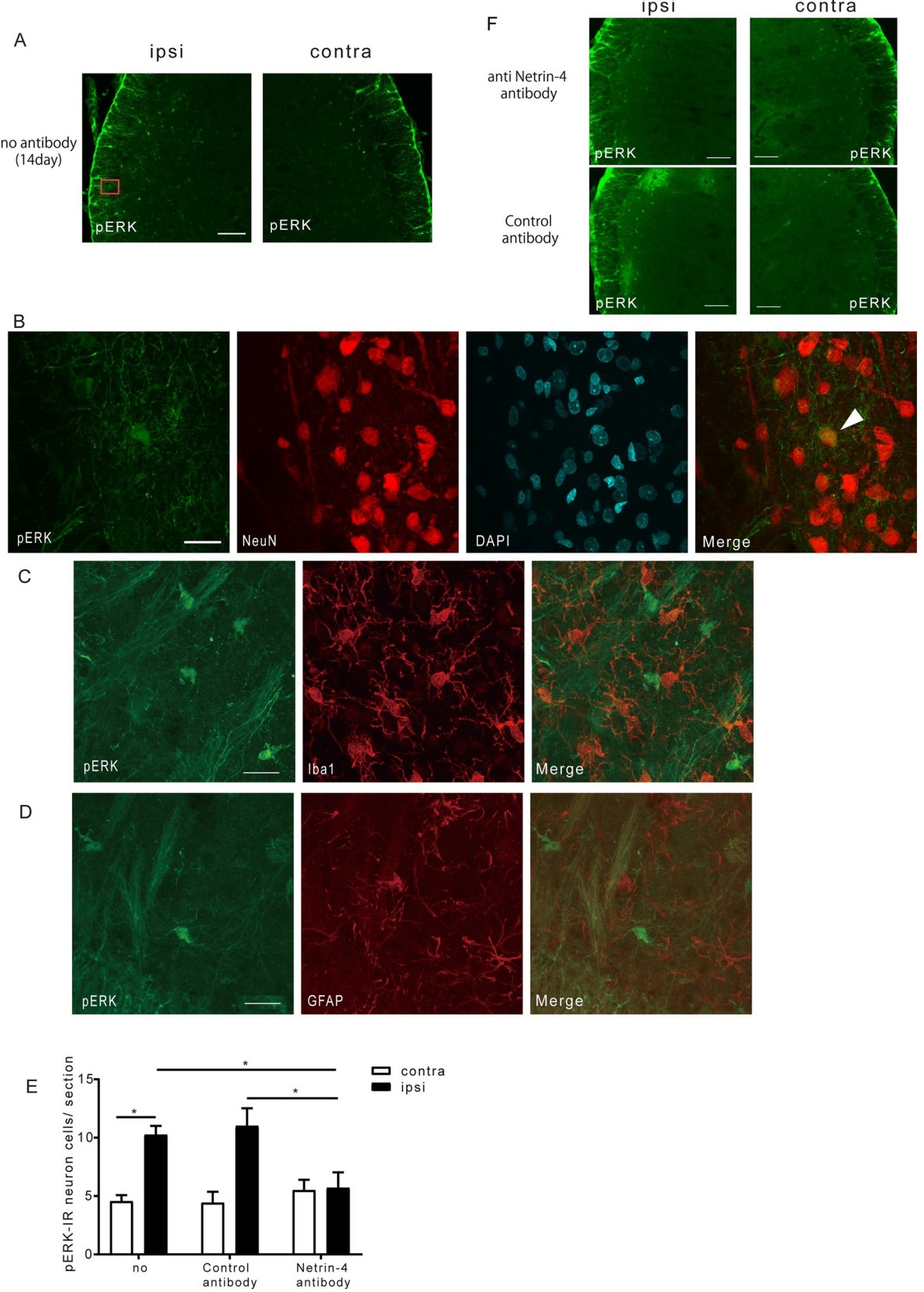

**Fig 3. Suppression of neuronal activation by administration of anti-Netrin-4 antibody.** (A) pERK immunofluorescence (green) staining in the Vc of ipsilateral and contralateral side. (B) pERK and NeuN (red) double immunofluorescence staining. The arrowhead indicates pERK positive neurons. (C and D) pERK (green) and Iba1or GFAP (red) double immunofluorescence staining. Bar: (A) 100 μm, (B, C, D) 20 μm. (E) The mean number of pERK-positive neurons in Vc-C1 after control or anti-Netrin-4 antibody administration (n = 4–6). The data are presented as the mean ± SEM. *, P<0.05; Tukey-Kramer test. (F) pERK immunofluorescence (green) staining in the Vc of ipsilateral and contralateral side. There were a number of pERK-positive sells in ipsilateral side after control antibody treatment, whereas anti-Netrin-4 antibody treatment reduced the number of them. Bar: 100 μm.

numerous studies [17, 19, 20]; moreover, it has been reported that chronic injury caused by peripheral nerve ligation induces allodynia or hyperalgesia to chemical, mechanical, and thermal stimuli [17, 21, 22]. In the established CCI-Sham injury model, infraorbital nerve ligation (CCI-injury) and non-ligation (Sham injury) were performed in the same rat. Numerous studies have analyzed the head-withdrawal threshold between the infraorbital nerve injury group (CCI-ION) and the pseudo injury group (Sham). A study reported a slightly decreased threshold on the injured and healthy side of the Sham and CCI-ION group, which recovered on 14 days after injury [19]. We observed recovery on the contralateral side and a significant decrease in the ipsilateral side at 14 days after injury. Therefore, to minimize the number of rats used in the experiment, we employed this method where the CCI-Sham injury was performed in the same rat.

The decreased escape thresholds caused by CCI-ION is observed from the first post-ligation day and lasts for at least 21 days [19, 20]. We observed the down threshold until 21 days after CCI-Sham injury. The hyperalgesia incidence on 14 days after CCI-Sham injury in our study was 30.4%, which is consistent with a previous study that employed a separate CCI and sham group [23].

Our findings indicated that allodynia developed on the ipsilateral side at 14 days after CCI-Sham injury and persisted until 21 days. Subsequent assessments of these rat models revealed a decreased head-withdrawal threshold at 14 days after injury.

## The mechanism of hyperalgesia and the role of Netrin-4

Ligation and inflammation could induce peripheral nerve injury. Inflammatory factors, including bradykinin and prostaglandin, decrease the threshold of Transient Receptor Potential Vanilloid 1 and increase Na+ channel excitability via individual receptors located on the peripheral side of the primary afferent nerve, which causes peripheral sensitization [24]. Subsequently, glutamate, Substance P, and brain-derived neurotrophic factor released from the central site of the primary afferent nerve bind to individual neuron receptors in the secondary afferent nerve. This is followed by MEK phosphorylation, which leads to ERK activation. This leads to the activation of the cAMP response element-binding protein in the neuron nucleus and expression of various genes, including c-Fos and Neurokinin 1 (NK-1) [25–27]. It has been reported that pERK triggers post-translational regulation of AMPA and NMDA receptor and contributes to central sensitization [28–30]. This sensitization is thought to lead to increased neuronal excitability, as well as allodynia and hyperalgesia occurrence. After spinal cord injury, there is temporary pERK expression at the surface of the spinal dorsal horn; subsequently, there is long-term pERK expression in microglia and astrocytes [31]. Direct pERK inhibition by a selective MEK inhibitor reduces formalin-induced pain behavior. Therefore, pERK is considered to be deeply involved in neuropathic development and maintenance [32, 33]. We observed a greater increase in pERK-positive neurons in the ipsilateral side than in the contralateral side at 14 days after CCI-Sham injury. This suggests that peripheral and central sensitization occurred at 14 days after injury and that neurons of the Vc contribute to hyperalgesia through pERK expression. However, the number of pERK-positive neurons in our study

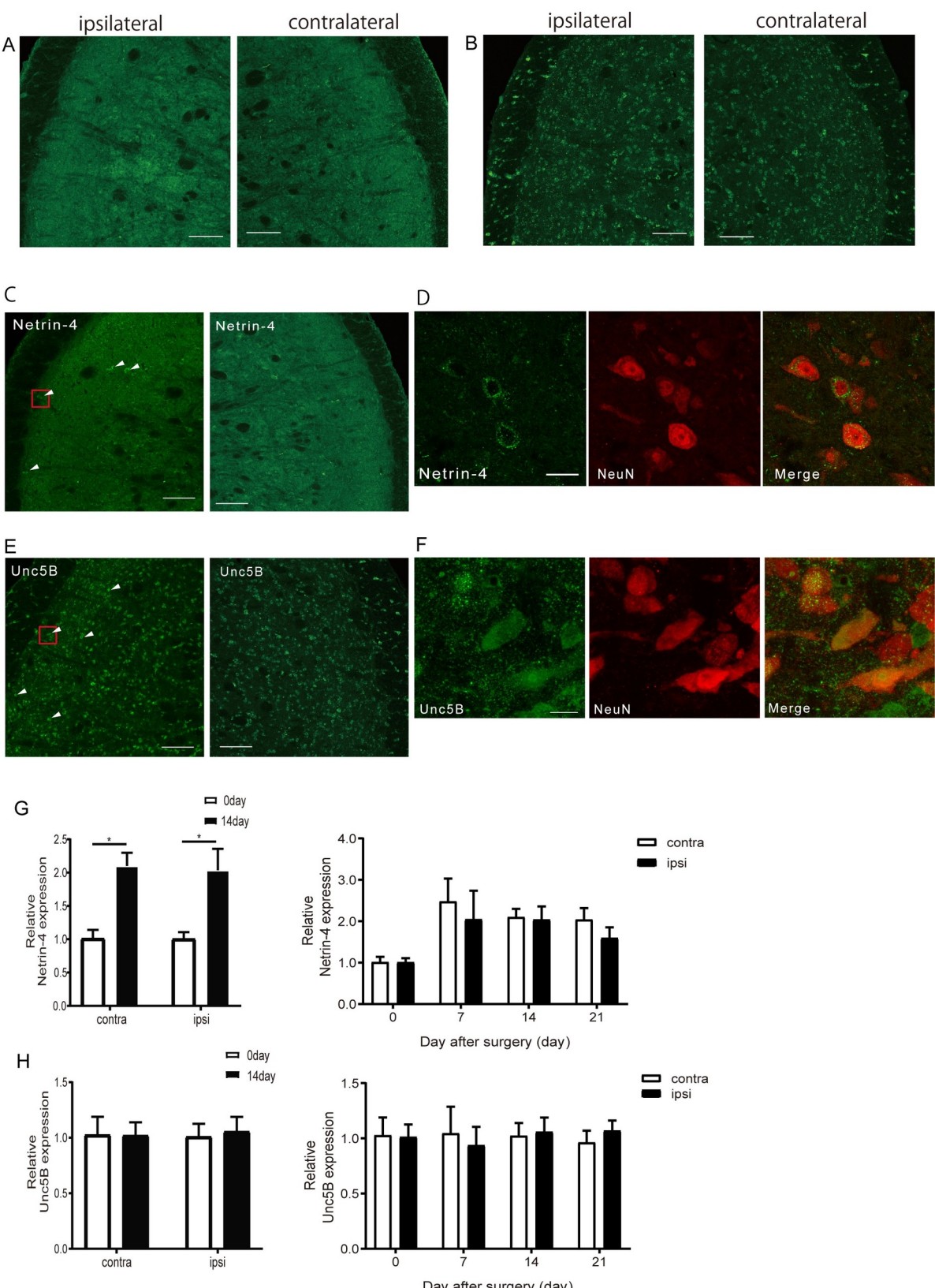

**Fig 4. Netrin-4 and Unc5B receptors are expressed in Vc-C1.** (A and B) Immunofluorescence staining for Netrin-4 (A) and Unc5B (B) in Vc of the ipsilateral side and contralateral side. Bar:100μm. (C and D) Netrin-4 (green) and NeuN (red) double immunofluorescence staining in Vc of the ipsilateral side. Bar: (C) 100μm (D) 10μm. (E and F) Unc5B (green) and NeuN (red) double immunofluorescence staining in Vc of the ipsilateral side. Bar: (E) 100μm (F) 10μm. (G and H) Netrin-4 (G) and Unc5B (F) mRNA expression in the ipsilateral (ipsi) and contralateral (contra) side at 14days after CCI-Sham injury (n = 3–5). The right graphs show the other day time results. The data are presented as the mean ± SEM. *, P<0.05; Mann-Whitney test.

was much less than that of previous reports [20, 34]. This could be partly attributed to differences in the mechanical stimulus intensity and stimulation method applied to the model. However, even with sciatic nerve or spinal cord injury induced by a slight stimulus, increased pERK expression has still been reported [35, 36]. Moreover, pERK expression has been reported to peak at several minutes after the noxious stimulus and to decrease by half after 10 minutes [32]. These findings suggest that differences in the number of cells could be attributed to differences in the time until tissue collection after mechanical stimulation. Further, there could have been reduced expression compared to that immediately after mechanical stimulation. Therefore, in the present study, if tissues had been collected sooner after mechanical stimulation, the number of pERK-positive neurons could have been confirmed to correspond to the observed pain behavior. Further, 30 minutes after mechanical stimulation, there was a small total number of pERK-positive neurons; nonetheless, it was significantly higher in the ipsilateral side than on the contralateral side. Moreover, sustained nerve cell activation may cause long-term neuropathic pain.

Contrastingly, upon spinal cord injury, SHP2 and NR2B, which are among the NMDA receptor subtypes, are activated when Netrin-4 secreted from interneurons in the lamina 2 of the dorsal horn binds to Unc5B [16]. Consequently, neurons are activated while neuropathic pain and inflammatory pain occurs. We observed that anti-Netrin-4 antibody administration through the cisterna magna increased the escape threshold by CCI-ION and reduced neuronal pERK expression in the Vc. However, Netrin-4 expression is increased also in the contralateral side at day 14. These findings suggest that Netrin-4 enhances hypersensitivity induced by damage to the infraorbital nerve, but that Netrin-4 is not sufficient to induce hypersensitivity. The molecular mechanism of the effect of Netrin-4 in the Vc might be explained by the previous observations that Netrin-4 binding to Unc5B leads to activation of SHP2 and the Ras-MAPK pathway [16, 37–43].

## The expression of Netrin-4 and Unc5B in Vc

Netrin-4 occurrence ranges widely in adult healthy brains in rats. It is not only expressed in the cerebral cortex but also the trigeminal spinal tract (Sp5) and Vi [14]. Therefore, Netrin-4 expression in the Vc is expected. Further, Unc5B has been reported as among the Netrin receptors that bind Netrin-1 or Netrin-4 [6, 13, 44].

Immunohistochemical analysis revealed Netrin-4 and Unc5B expression in the Vc~C1 at 14 days after CCI-Sham injury which was the day of hyperalgesia development. Moreover, quantitative mRNA analysis of the Vc~C1 indicated significantly increased Netrin-4 at 14 days after CCI-Sham injury compared with before injury. This indicates that hyperalgesia is caused by increased netrin-4 secretion, which triggers neuronal activation. In our study, although the number of Netrin-4-positive cells in the Vc appear to be a few, it may reflect the sensitivity of the antibodies employed. As Netrin-4 is a secreted protein, it would be difficult to detect the low amount of the Netrin-4 protein by immunohistochemistry. It should also be noted that the mechanism of upregulation of Netrin-4 in the Vc region remains a question and should be explored in the future.

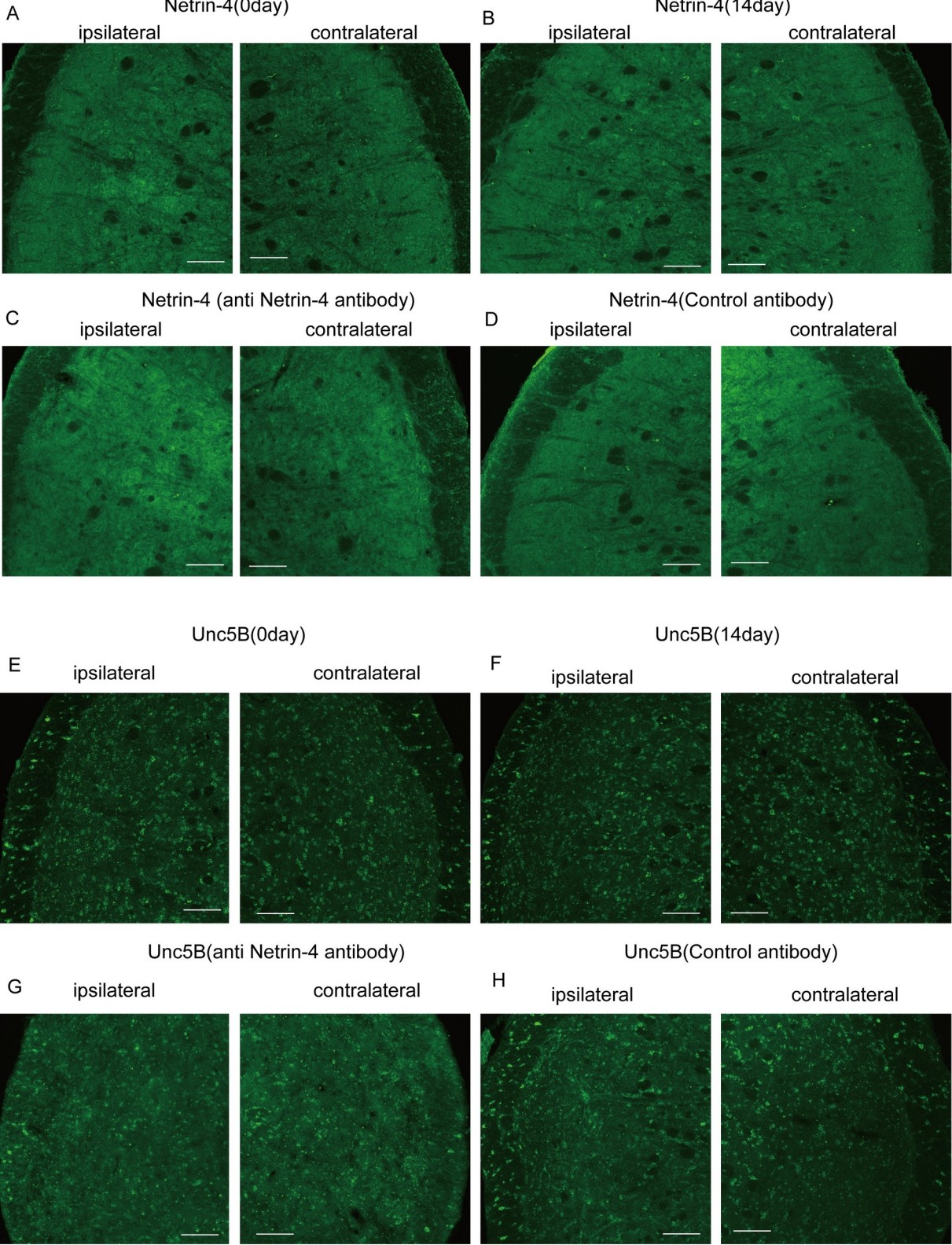

**Fig 5. Expression of Netrin-4 and Unc5B receptor inVc-C1.** (A and B) Netrin-4 expression in Vc of ipsilateral and contralateral side at 0 and 14days. (C and D) Netrin-4 expression in Vc of ipsilateral and contralateral side after anti-Netrin-4 antibody or Control antibody treatment. (E and F) Unc5B expression in Vc of ipsilateral and contralateral side at 0 and 14days. (G and H) Unc5B expression in Vc of ipsilateral and contralateral side after anti-Netrin-4 antibody or Control antibody treatment. Bars: 100μm.

Contrastingly, there was no change in Unc5B mRNA expression in the Vc although it was increased in the spinal cord after nerve injury. This could be attributed to differences between the brain and spinal cord; however, it has been reported that Unc5B mRNA expression decreased and recovered after spinal cord injury [45]. Therefore, Unc5B may show different changes at other time points. Further, it is possible that there are changes in Unc5B protein levels at 14 days after CCI-Sham injury. Therefore, there is a need for further studies. Furthermore, Netrin-4 has been reported to bind to laminin-γ1[46, 47]. Therefore, there is a need to consider the involvement of receptors other than Unc5B.

Moreover, there was no significant difference in mRNA Netrin-4 and Unc5B expression in the Vc between the ipsilateral side and contralateral side despite these differences being observed in the spinal cord. This suggests that SHP2 and NR2B activation change at the ipsilateral side of the Vc similar to in the spinal cord [16]. There is a need for further analysis to elucidate the underlying mechanism.

## The analgesic effects of anti-Netrin-4 antibody

We observed that the administration of the anti-Netrin-4 antibody through the cisterna magna increased the head-withdrawal threshold for mechanical stimulation and suppressed neuronal activation in the Vc. These findings suggest that the anti-Netrin-4 antibody presents an analgesic effect by inhibiting the binding of Netrin-4 to Unc5B.

In adult humans, Netrin-4 is present in the heart, kidney, and spleen; further, it is slightly expressed in the brain and spinal cord [7, 48]. The presence of Netrin-4 widely varies in adult healthy brains in rats [14]. There is no abnormality in nerve circuit formation, as well as reproduction and organ formation, in the developmental stage of Netrin-4 knockout rats [16]. The anti-Netrin-4 antibody used in our study has been shown to reduce neuropathic pain caused by infraorbital nerve injury. Moreover, it did not induce significant weight loss or death upon administration. These findings suggest that the anti-Netrin-4 antibody is relative safety in animals and could be applied to humans.

Our findings suggest the following: CCI-ION induced persistent trigeminal nerve excitement and increased Netrin-4 secretion. Subsequently, hyperalgesia was induced by increased Netrin-4 binding to Unc5B. Other than suppressing neuronal activity, anti-Netrin-4 antibody is effective for pain hypersensitivity due to neuropathic pain and could be used in humans in the future.

## Author Contributions

**Conceptualization:** Yuka Honjo, Yuki Fujita, Toshihide Yamashita.

**Data curation:** Yuka Honjo, Yuki Fujita.

**Formal analysis:** Yuka Honjo, Yuki Fujita.

**Investigation:** Yuka Honjo, Yuki Fujita.

**Methodology:** Yuka Honjo, Yuki Fujita.

**Supervision:** Yuki Fujita, Hitoshi Niwa, Toshihide Yamashita.

**Validation:** Yuki Fujita.

**Writing – original draft:** Yuka Honjo.

**Writing – review & editing:** Yuki Fujita, Toshihide Yamashita.

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
