## [Decision Letter · Decision Letter 0]

6 Oct 2020

PONE-D-20-28596

Increased expression of Netrin-4 is associated with allodynia in a trigeminal neuropathic pain model rats by infraorbital nerve injury

PLOS ONE

Dear Dr. Yamashita,

Thank you for submitting your manuscript to PLOS ONE. After careful consideration, we feel that it has merit but does not fully meet PLOS ONE’s publication criteria as it currently stands. Therefore, we invite you to submit a revised version of the manuscript that addresses the points raised during the review process.

ACADEMIC EDITOR: Your manuscript has been assessed by the reviewers. Although it is of interest, we are unable to consider it for publication in its current form. The reviewers have raised several points which we believe would improve the manuscript. 

We look forward to receiving your revised manuscript.

Kind regards,

Masabumi Minami, Ph.D.

Academic Editor

PLOS ONE

Journal Requirements:

2) Please include captions for your Supporting Information files at the end of your manuscript, and update any in-text citations to match accordingly. Please see our Supporting Information guidelines for more information: http://journals.plos.org/plosone/s/supporting-information.

Reviewers' comments:

Reviewer's Responses to Questions

**Comments to the Author**

1. Is the manuscript technically sound, and do the data support the conclusions?

Reviewer #1: Yes

Reviewer #2: Yes

2. Has the statistical analysis been performed appropriately and rigorously? 

Reviewer #1: Yes

Reviewer #2: Yes

3. Have the authors made all data underlying the findings in their manuscript fully available?

Reviewer #1: Yes

Reviewer #2: Yes

4. Is the manuscript presented in an intelligible fashion and written in standard English?

Reviewer #1: Yes

Reviewer #2: Yes

5. Review Comments to the Author

Reviewer #1: Honjo et al. report the contribution of Netrin-4 to mechanical hypersensitivity in a rat model of trigeminal neuropathic pain. This study is interesting, but I have a few issues that should be addressed.

Major comments

1. The role of Netrin-4 was investigated using its antibody. The authors previously demonstrated its role in neuropathic pain using Netrin-4-mutant rats. Why do they use the mutant rats in this study? If Netrin-4-mutant rats exhibit reduction of behavioral hypersensitivity after infraorbital nerve injury, the authors’ conclusion of this study would be strengthened.

2. Phosphorylated-ERK has been reported to be seen primarily in glial cells (not neurons) 14 days after nerve injury (PMID: 15733640). These cells are crucial for neuronal sensitization associated with neuropathic pain (PMID: 19741123). Did the authors detect p-ERK in astrocytes or microglia in the Vc after nerve injury?

3. Netrin-4 expression is also increased in the contralateral side at day 14 (Figure 4). However, pain threshold in the contralateral side at that time was normal (Figure 1). These data imply that Netrin-4 is not sufficient to mechanical hypersensitivity. On the other hand, the authors previously reported that intrathecal administration of Netrin-4 to normal rats induces mechanical hypersensitivity. How do the authors reconcile the discrepancy? This should be discussed.

Minor comment

1. (Line 258-260) The authors might mistakenly cite Refs #30 and 31 (these papers do not use a spinal cord injury model).

Reviewer #2: The authors of this work have shown that the expression of Netrin-4 is increased in the Vc 14 days after ligation of infraorbital nerve, and thereby activation of Netrin-4/Unc5B signaling axis are responsible for the development of neuropathic pain after ION-CCI. Each result may not be novel, but the hypotheses of this study have been theoretically tested and discussed, and the manuscript is well-written. However, in the current format, the clarity of these findings should not be enough. There are several points that the authors should further address to their excellent manuscript.

Major Comment

1. Throughout the manuscript, there seems to be a lack of explanation of the Netrin-4/Unc5B signaling axis in the Vc region that are activated after ION-CCI. For example, it is unclear how Netrin-4 is up-regulated in the VC region after nerve injury. It is also difficult to understand how the authors conclude that Netrin-4 is upregulated in interneurons IN THE VC region. Furthermore, it would also be necessary to mention through what signaling pathway ERK is activated after Netrin-4 binds to Unc5B.

2. The authors argued that Netrin-4 may participate in the sensitization process leading to neuropathic pain. Of course, their study using anti-Netrin-4 antibody highlight pro-nociceptive roles of Netrin-4. However, based on the series of there findings, it is not sufficient to conclude that netrin-4 is involved in central sensitization. For example, their conclusions would be reinforced by showing the time course of changes in netrin-4 expression after ION-CCI.

3. In order to conclude the nociceptive roles of Netrin-4 in the Vc, I strongly recommend not only to experiment with neutralizing Netrin-4 antibody, but also to investigate whether the administration of recombinant Netrin-4 protein mimics pain behaviors and activation of ERK in the Vc.

4. My major concern with their data is that complete lack of staining images in the figures. For example, in figure 3, immunohistochemical images of contralateral side and images in the netrin-4 neutralizing antibody treatment group are absolutely necessary. Similarly, staining images of contralateral and ipsilateral side on day 0 and 14 MUST be included in figure 4.

5. Although the authors have proposed a role for netrin-4 in the development of pain, netrin-4-positive cells appear to be a few in the Vc even after ION-CCI. This discrepancy needs to be explained in the manuscript.

Minor Comment

1. P.11, line 215: Is the sentence "we performed quantitative mRNA analysis by RT-PCR." correct? I think they conducted qPCR analysis in the study, and should be corrected.

6. PLOS authors have the option to publish the peer review history of their article (what does this mean?). If published, this will include your full peer review and any attached files.

Reviewer #1: No

Reviewer #2: No

---

## [Author Response · Author response to Decision Letter 0]

25 Mar 2021

Reviewer #1: 

Major comments

1. The role of Netrin-4 was investigated using its antibody. The authors previously demonstrated its role in neuropathic pain using Netrin-4-mutant rats. Why do they use the mutant rats in this study? If Netrin-4-mutant rats exhibit reduction of behavioral hypersensitivity after infraorbital nerve injury, the authors’ conclusion of this study would be strengthened.

Netrin-4 mutant rats show developmental abnormality of the thalamocortical axon branching (ref. 13 in the manuscript). Therefore, if hyperalgesia does not occur in the maxillofacial region after CCI-ION in the mutant rats, we should consider the possibility that the abnormal wiring of the sensory neural pathway may cause this effect. In addition, some previous reports (References [1-3] in this letter) show that Netrin-4 mutant animals show abnormal vascular structure in retina and in blood vessels in the trigeminal nerve area. In the present study, we intended to assess if transient suppression of Netrin-4 was effective in suppressing hypersensitivity after infraorbital nerve injury. Practically, Netrin-4-mutant rats are not currently available due to reconstruction of the animal testing facility in Osaka University. Therefore, we decided not to perform the experiments using Netrin-4 mutant rats.

2. Phosphorylated-ERK has been reported to be seen primarily in glial cells (not neurons) 14 days after nerve injury (PMID: 15733640). These cells are crucial for neuronal sensitization associated with neuropathic pain (PMID: 19741123). Did the authors detect p-ERK in astrocytes or microglia in the Vc after nerve injury?

We added images that show immunohistochemical staining for pERK in astrocytes and microglia (Figure 3C, D). However, we could not detect pERK signals in astrocyte or microglia (page 10, last paragraph).

3. Netrin-4 expression is also increased in the contralateral side at day 14 (Figure 4). However, pain threshold in the contralateral side at that time was normal (Figure 1). These data imply that Netrin-4 is not sufficient to mechanical hypersensitivity. On the other hand, the authors previously reported that intrathecal administration of Netrin-4 to normal rats induces mechanical hypersensitivity. How do the authors reconcile the discrepancy? This should be discussed.

We carried out additional experiments to address the point raised by the reviewer. We examined expression of Netrin-4 and Unc5B at days 0 and 14 (Figure 5A, B, E, F), and after anti-Netrin-4 antibody or control antibody treatment (Figure 5C, D, G, H). We did not observe the difference of the expression of Netrin-4 between ipsilateral and contralateral side in each group examined (Figure 5A-H). 

Then, we tried administration of Netrin-4 from cisterna magna of rats, but could not observe mechanical hypersensitivity in the area of infraorbital nerve (data not shown). From these observations, we considered that Netrin-4 may enhance hypersensitivity following damage to the infraorbital nerve, but that Netrin-4 is not sufficient to induce hypersensitivity. 

We stated in the text as follow (page 14, last paragraph ~ page 15, first paragraph). “We observed that anti-Netrin-4 antibody administration through the cisterna magna increased the escape threshold by CCI-ION and reduced neuronal pERK expression in the Vc. However, Netrin-4 expression is increased also in the contralateral side at day 14. These findings suggest that Netrin-4 enhances hypersensitivity induced by damage to the infraorbital nerve, but that Netrin-4 is not sufficient to induce hypersensitivity. The molecular mechanism of the effect of Netrin-4 in the Vc might be explained by the previous observations that Netrin-4 binding to Unc5B leads to activation of SHP2 and the Ras-MAPK pathway [16, 37-43] .”

”

Minor comment

1. (Line 258-260) The authors might mistakenly cite Refs #30 and 31 (these papers do not use a spinal cord injury model).

We corrected those references.

 

Reviewer #2: 

Major Comment

1. Throughout the manuscript, there seems to be a lack of explanation of the Netrin-4/Unc5B signaling axis in the Vc region that are activated after ION-CCI. For example, it is unclear how Netrin-4 is up-regulated in the VC region after nerve injury. It is also difficult to understand how the authors conclude that Netrin-4 is upregulated in interneurons in the VC region. Furthermore, it would also be necessary to mention through what signaling pathway ERK is activated after Netrin-4 binds to Unc5B.

The mechanism of upregulation of Netrin-4 in the Vc region remains a question. We added the description (page 15, third paragraph).

Regarding the cell types, we could not conclude that Netrin-4 was upregulated in interneurons in the VC region. Therefore, we did not mention that Netrin-4 was upregulated in interneurons in the VC region. In the CCI model, we identified that the central cells in the spinal cord were the source of Netrin-4 (ref 16 in the manuscript). However, the current knowledge about the interneurons in the Vc is sparce and we could not determine the exact cell type that expressed Netrin-4 in the present study. 

Regarding the signaling pathway, we added discussion (page 14, last paragraph ~ page 15, first paragraph). “We observed that anti-Netrin-4 antibody administration through the cisterna magna increased the escape threshold by CCI-ION and reduced neuronal pERK expression in the Vc. However, Netrin-4 expression is increased also in the contralateral side at day 14. These findings suggest that Netrin-4 enhances hypersensitivity induced by damage to the infraorbital nerve, but that Netrin-4 is not sufficient to induce hypersensitivity. The molecular mechanism of the effect of Netrin-4 in the Vc might be explained by the previous observations that Netrin-4 binding to Unc5B leads to activation of SHP2 and the Ras-MAPK pathway [16, 37-43] .”

2. The authors argued that Netrin-4 may participate in the sensitization process leading to neuropathic pain. Of course, their study using anti-Netrin-4 antibody highlight pro-nociceptive roles of Netrin-4. However, based on the series of there findings, it is not sufficient to conclude that netrin-4 is involved in central sensitization. For example, their conclusions would be reinforced by showing the time course of changes in netrin-4 expression after ION-CCI.

We analyzed the time course changes of netrin-4 and Unc5B expression after ION-CCI by real-time PCR. Compared with pre-injury levels, there was a significant increase in the Netrin-4, but not Unc5b, levels from 7 to 21 days after CCI-Sham injury (Figure 4G, H).

3. In order to conclude the nociceptive roles of Netrin-4 in the Vc, I strongly recommend not only to experiment with neutralizing Netrin-4 antibody, but also to investigate whether the administration of recombinant Netrin-4 protein mimics pain behaviors and activation of ERK in the Vc.

We previously reported that intrathecal administration of Netrin-4 to normal rats’ spinal cord induced mechanical hypersensitivity (ref 16 in the manuscript). In order to address the point raised by the reviewer, we tried administration of Netrin-4 from cisterna magna of rats, but could not observe mechanical hypersensitivity in the area of infraorbital nerve (data not shown). From these data, we considered that Netrin-4 may enhance hypersensitivity following damage to the infraorbital nerve, but that Netrin-4 is not sufficient to induce hypersensitivity (page 14, last paragraph ~ page 15, first paragraph).

4. My major concern with their data is that complete lack of staining images in the figures. For example, in figure 3, immunohistochemical images of contralateral side and images in the netrin-4 neutralizing antibody treatment group are absolutely necessary. Similarly, staining images of contralateral and ipsilateral side on day 0 and 14 MUST be included in figure 4.

We examined expression of Netrin-4 and Unc5B at days 0 and 14 (Figure 5A, B, E, F), and after anti-Netrin-4 antibody or control antibody treatment (Figure 5C, D, G, H). We did not observe the difference of the expression of Netrin-4 between ipsilateral and contralateral side in each group examined (Figure 5A-H). Moreover, we added the images for pERK staining with or without anti-Netrin-4 antibody treatment (Figure 3F)

5. Although the authors have proposed a role for netrin-4 in the development of pain, netrin-4-positive cells appear to be a few in the Vc even after ION-CCI. This discrepancy needs to be explained in the manuscript.

In our study, although the number of Netrin-4-positive cells in the Vc appear to be a few, it may reflect the sensitivity of the antibodies employed. As Netrin-4 is a secreted protein, it would be difficult to detect the low amount of the Netrin-4 protein by immunohistochemistry. We added the description in Discussion (page 15, second paragraph).

Minor Comment

1. P.11, line 215: Is the sentence "we performed quantitative mRNA analysis by RT-PCR." correct? I think they conducted qPCR analysis in the study, and should be corrected.

We corrected the text. (P.11, first paragraph “we performed quantitative mRNA analysis by RT-PCR.”→ “we performed quantitative mRNA analysis by real-time PCR.”)

1. Crespo-Garcia S, Reichhart N, Wigdahl J, Skosyrski S, Kociok N, Strauß O, et al. Lack of netrin-4 alters vascular remodeling in the retina. Graefes Arch Clin Exp Ophthalmol. 2019;257(10):2179-84. Epub 2019/08/28. doi: 10.1007/s00417-019-04447-3. PubMed PMID: 31451908.

2. Crespo-Garcia S, Reichhart N, Wigdahl J, Skosyrski S, Kociok N, Strauß O, et al. Correction to: Lack of netrin-4 alters vascular remodeling in the retina. Graefes Arch Clin Exp Ophthalmol. 258. Germany2020. p. 217.

3. Kociok N, Crespo-Garcia S, Liang Y, Klein SV, Nürnberg C, Reichhart N, et al. Lack of netrin-4 modulates pathologic neovascularization in the eye. Sci Rep. 2016;6:18828. Epub 2016/01/07. doi: 10.1038/srep18828. PubMed PMID: 26732856; PubMed Central PMCID: PMCPMC4702134.

4. Hayano Y, Takasu K, Koyama Y, Yamada M, Ogawa K, Minami K, et al. Dorsal horn interneuron-derived Netrin-4 contributes to spinal sensitization in chronic pain via Unc5B. Journal of Experimental Medicine. 2016;213(13):2949-66.

5. Tong J, Killeen M, Steven R, Binns KL, Culotti J, Pawson T. Netrin stimulates tyrosine phosphorylation of the UNC-5 family of netrin receptors and induces Shp2 binding to the RCM cytodomain. J Biol Chem. 2001;276(44):40917-25. Epub 2001/09/05. doi: 10.1074/jbc.M103872200. PubMed PMID: 11533026.

6. Krapivinsky G, Krapivinsky L, Manasian Y, Ivanov A, Tyzio R, Pellegrino C, et al. The NMDA receptor is coupled to the ERK pathway by a direct interaction between NR2B and RasGRF1. Neuron. 2003;40(4):775-84. Epub 2003/11/19. doi: 10.1016/s0896-6273(03)00645-7. PubMed PMID: 14622581.

7. Kim MJ, Dunah AW, Wang YT, Sheng M. Differential roles of NR2A- and NR2B-containing NMDA receptors in Ras-ERK signaling and AMPA receptor trafficking. Neuron. 2005;46(5):745-60. Epub 2005/06/01. doi: 10.1016/j.neuron.2005.04.031. PubMed PMID: 15924861.

8. Easton JB, Royer AR, Middlemas DS. The protein tyrosine phosphatase, Shp2, is required for the complete activation of the RAS/MAPK pathway by brain-derived neurotrophic factor. J Neurochem. 2006;97(3):834-45. Epub 2006/04/01. doi: 10.1111/j.1471-4159.2006.03789.x. PubMed PMID: 16573649.

9. Round J, Stein E. Netrin signaling leading to directed growth cone steering. Curr Opin Neurobiol. 2007;17(1):15-21. Epub 2007/01/27. doi: 10.1016/j.conb.2007.01.003. PubMed PMID: 17254765.

10. Peng HY, Chen GD, Lai CY, Hsieh MC, Lin TB. Spinal SIRPα1-SHP2 interaction regulates spinal nerve ligation-induced neuropathic pain via PSD-95-dependent NR2B activation in rats. Pain. 2012;153(5):1042-53. Epub 2012/03/20. doi: 10.1016/j.pain.2012.02.006. PubMed PMID: 22425446.

11. Abbasi M, Gupta V, Chitranshi N, You Y, Dheer Y, Mirzaei M, et al. Regulation of Brain-Derived Neurotrophic Factor and Growth Factor Signaling Pathways by Tyrosine Phosphatase Shp2 in the Retina: A Brief Review. Front Cell Neurosci. 2018;12:85. Epub 2018/04/11. doi: 10.3389/fncel.2018.00085. PubMed PMID: 29636665; PubMed Central PMCID: PMCPMC5880906.

---

## [Decision Letter · Decision Letter 1]

19 Apr 2021

Increased expression of Netrin-4 is associated with allodynia in a trigeminal neuropathic pain model rats by infraorbital nerve injury

PONE-D-20-28596R1

Dear Dr. Yamashita,

We’re pleased to inform you that your manuscript has been judged scientifically suitable for publication and will be formally accepted for publication once it meets all outstanding technical requirements.

Kind regards,

Masabumi Minami, Ph.D.

Academic Editor

PLOS ONE

Additional Editor Comments (optional):

Reviewers' comments:

Reviewer's Responses to Questions

**Comments to the Author**

1. If the authors have adequately addressed your comments raised in a previous round of review and you feel that this manuscript is now acceptable for publication, you may indicate that here to bypass the “Comments to the Author” section, enter your conflict of interest statement in the “Confidential to Editor” section, and submit your "Accept" recommendation.

Reviewer #1: All comments have been addressed

Reviewer #2: All comments have been addressed

2. Is the manuscript technically sound, and do the data support the conclusions?

Reviewer #1: Yes

Reviewer #2: Yes

3. Has the statistical analysis been performed appropriately and rigorously? 

Reviewer #1: Yes

Reviewer #2: Yes

4. Have the authors made all data underlying the findings in their manuscript fully available?

Reviewer #1: Yes

Reviewer #2: No

5. Is the manuscript presented in an intelligible fashion and written in standard English?

Reviewer #1: Yes

Reviewer #2: Yes

6. Review Comments to the Author

Reviewer #1: (No Response)

Reviewer #2: The authors have responded to the questions posed in the previous manuscript with appropriate answers and additional experiments, and I feel the manuscript has been greatly improved. I have no further questions.

7. PLOS authors have the option to publish the peer review history of their article (what does this mean?). If published, this will include your full peer review and any attached files.

Reviewer #1: No

Reviewer #2: No

---

## [Editor Report · Acceptance letter]

21 Apr 2021

PONE-D-20-28596R1 

Increased expression of Netrin-4 is associated with allodynia in a trigeminal neuropathic pain model rats by infraorbital nerve injury 

Dear Dr. Yamashita:

I'm pleased to inform you that your manuscript has been deemed suitable for publication in PLOS ONE. Congratulations! Your manuscript is now with our production department. 

Kind regards, 

on behalf of

Dr. Masabumi Minami 

Academic Editor

PLOS ONE